# How Empowering Leadership Drives Proactivity in the Chinese IT Industry: Mediation Through Team Job Crafting and Psychological Safety with ICT Knowledge as a Moderator

**DOI:** 10.3390/bs15050609

**Published:** 2025-05-01

**Authors:** Juanxiu Piao, Juhee Hahn

**Affiliations:** 1The Graduate School, Chung-Ang University, Seoul 06974, Republic of Korea; parkstyle7@cau.ac.kr; 2Department of Business Management, Chung-Ang University, Seoul 06974, Republic of Korea

**Keywords:** empowering leadership, workplace proactivity, team job crafting, psychological safety, access to knowledge via ICT, job demands–resources (JD-R) theory, hierarchical analyses, China

## Abstract

In China’s rapidly digitizing IT industry, empowering leadership has become a crucial catalyst for workplace proactivity; however, the mechanisms linking leadership practices to individual proactive behaviors remain underexplored. This study addresses this gap by proposing a multi-level framework that integrates team processes and technological contexts. Based on the job demands–resources theory, the research examines the mechanisms of empowering leadership through parallel team-level pathways and the influence of digital infrastructure on these dynamics. Data were gathered in three phases from 510 employees across 74 teams in seven IT firms. Hierarchical analyses with SPSS 27.0, AMOS 28.0, and HLM 6.08 revealed three pathways: empowering leadership significantly enhances workplace proactivity, with team job crafting and psychological safety serving as sequential mediators. Moreover, access to knowledge via ICT moderates the relationship between team job crafting and workplace proactivity. This study theoretically contests sequential mediation assumptions by demonstrating parallel, non-overlapping mechanisms and redefines ICT’s role as a contextual enhancer in digital workplaces. Practically, it offers organizations a modular strategy: leaders can prioritize either job crafting systems or psychological safety climates to foster proactivity, depending on their team’s technological readiness. These insights offer practical recommendations for optimizing leadership practices in high-pressure IT environments, where digital tools and team dynamics influence employee initiative.

## 1. Introduction

As digital transformation and flexible working models proliferate, organizations increasingly prioritize employee motivation, engagement, mental health, and operational efficiency ([15]). In the modern digital age, information and communication technology (ICT) holds a pivotal role in the workplace. Most organizations rely on these technologies to enhance efficiency, foster innovation, and adapt to the evolving market landscape ([5]). The extensive adoption of ICT has revolutionized work practices and redefined interactions between leaders and employees, thereby facilitating the emergence of digital leadership. Organizations are progressively embracing innovative work models to adapt to the agility and dynamism of emerging technologies, artificial intelligence, and digitalization ([5]).

China’s IT industry offers a distinctive framework for exploring the relationship between empowering leadership and employee proactivity. As the world’s second-largest digital economy ([12]), it embodies a notable paradox in the practice of digital-era leadership. Despite leading globally in ICT adoption, over 70% of IT employees in China reportedly experience chronic overwork under the “996” schedule (9 a.m.–9 p.m., six days a week), a practice institutionalized in firms such as Tencent and Alibaba ([16]). This contrasts with Western technology sectors, where employee autonomy is frequently linked to reduced burnout and enhanced well-being ([27]).

This divergence indicates that the impact of empowering leadership may be profoundly influenced by cultural and organizational contexts. For instance, Huawei’s “Iron Triangle” model illustrates that authority is delegated only after hierarchical roles are explicitly defined ([37]). This reflects a prevalent trend among Chinese firms to combine empowerment with structured control mechanisms, highlighting the need to adapt Western leadership constructs to China’s high-intensity, culturally nuanced IT landscape.

In a complex and volatile market, the emphasis on enhancing organizational agility increasingly depends on employee proactivity in the workplace ([10]). Previous studies demonstrate that individuals with proactive personalities exhibit elevated career success and job performance ([64]). Proactivity is a critical catalyst for organizational agility, particularly in volatile markets. This change-oriented behavior mitigates potential issues, diminishes uncertainties, fosters innovative ideas within the organization, and capitalizes on future opportunities. Consequently, it enhances the efficiency of both individuals and organizations ([21]). The significant impact of exploring the determinants of workplace proactivity renders it essential ([24]).

Empowering leadership is extensively promoted to enhance proactivity ([63]); however, its implementation in China’s IT sector reveals a critical tension between autonomy and structured control. It establishes an environment where employees are empowered to address challenging tasks autonomously, without the need for continual leader intervention ([62]). Investigating how empowering leadership fosters workplace proactivity among employees in Chinese IT firms and its underlying mechanisms enhances our comprehension of leadership’s impact on employee behavior in the contemporary economic landscape. This understanding is essential for attaining organizational success in a competitive and dynamic market.

Recent studies have increasingly emphasized team-based work as a manifestation of organizational behavior rather than individual task execution ([46]). As digitalization advances, employees increasingly depend on digital platforms for their daily activities, frequently utilizing ICT ([62]). ICT can enhance team productivity and stimulate active participation among team members via real-time knowledge sharing and collaboration platforms. When team members autonomously modify and customize their work tasks, this proactive behavior can significantly impact workplace proactivity ([66]). Furthermore, psychological safety within organizations is essential. Team members exhibit heightened proactivity and innovation when they perceive psychological safety ([2]). Employees can demonstrate proactive behavior by voluntarily proposing ideas, collaborating in teams, embracing new challenges, and enhancing workplace proactivity ([23]). A meta-analysis conducted by [26] ([26]) reveals that the beneficial impact of psychological safety, as a fundamental organizational behavioral variable, is universally relevant across diverse cultures and industries. This discovery indicates that establishing a psychologically safe environment is essential for enhancing team learning and fostering proactive employee behaviors, relevant in both developed Western markets and the rapidly expanding Chinese IT industry. Nonetheless, critical gaps persist. Although ICT facilitates knowledge sharing ([62]), its potential to amplify job demands—such as information overload—in empowering leadership contexts remains empirically unexamined. These omissions limit our ability to predict the success or failure of empowering leadership in digitally intensive work environments. [54] ([54]) observed that both individuals and organizations benefit from well-designed work; however, the specific causality remains ambiguous. Therefore, this study explores the impact of team job crafting on individual psychological safety. Utilizing ICT for knowledge acquisition facilitates the identification of key factors that foster employee proactivity in the workplace, thereby enabling organizations to establish a digitally supportive work environment that promotes learning, collaboration, and innovation. Organizations can enhance employee proactivity by cultivating a culture of proactive learning in a digital context ([55]).

This study explores the impact of enhanced leadership capabilities in Chinese IT firms during the digital age on employee workplace proactivity. It aims to address the following unresolved questions regarding China’s digitally intensive IT sector:

Q1: How do team job crafting and psychological safety jointly mediate the relationship between empowering leadership and employee proactivity within Chinese IT teams?

Q2: Under what conditions does ICT-enabled knowledge acquisition capability enhance or diminish the relationship between team job crafting and employee proactivity?

Q3: How can an enhanced job demands–resources (JD-R) framework provide evidence-based thresholds for balancing autonomy and employee motivation in high-pressure digital contexts?

This study investigates these questions to enhance leadership theory in digitally transformed workplaces, highlights the dynamic interplay between leadership practices and ICT infrastructure, and provides practical insights for promoting sustainable proactivity in fast-paced, knowledge-intensive industries.

## 2. Theoretical Foundation and Hypotheses Development

### 2.1. Empowering Leadership and Workplace Proactivity

Empowering leadership is characterized by leaders delegating authority to employees, fostering autonomous decision-making, and encouraging self-directed initiatives. This leadership style entails delegating power, promoting self-management, and providing employees with the autonomy and confidence to independently manage complex tasks. It involves establishing an environment where employees are empowered to address complex tasks independently without continual leader intervention ([1]; [63]). Empowered employees experience greater comfort and diminished job constraints, thereby enhancing their propensity to assist others and proactively engage in tasks. Furthermore, the heightened sense of identification with their roles motivates them to contribute positively to the organization ([7]). When individuals regard roles as significant, they are more inclined to actively seek information from diverse sources and exert increased effort in thoughtfully resolving problems ([73]). Employees tend to exhibit proactive behavior when significantly motivated by high empowerment and autonomy. For instance, empowerment motivates employees to openly share innovative ideas and actively engage in change-oriented behaviors ([6]).

Workplace proactivity denotes employees’ capacity to undertake self-initiated measures to foresee changes in their work and respond to future opportunities rather than passively experiencing developments ([73]). In workplace proactivity, empowering leadership is critical, especially when employees must exercise considerable discretion in their tasks ([63]). Empowering leadership fosters proactive behavior in the contemporary industry ([4]). Empowering leadership inspires employees to assume responsibility, confront challenges, and collaborate effectively ([38]). In this context, empowering leadership fosters employees’ initiative and independent workplace behavior. This may result in employees exercising autonomy in decision-making rather than being influenced by their leaders ([4]). Leaders’ persistent endeavors to empower employees significantly impacts proactive work behavior ([70]). Empowering leadership cultivates employees’ sense of responsibility and autonomy by prioritizing trust, empowerment, and support. This fosters a greater inclination for engaging in proactive and innovative work initiatives. This leadership strategy enhances employees’ work engagement, satisfaction, and overall performance ([4]; [52]). In the competitive Chinese IT industry, work demands are frequently elevated, necessitating adequate work resources, such as empowering leadership, to alleviate this pressure. Based on JDR theory, this study hypothesizes that empowering leadership can be a significant work resource that fosters proactive employee behaviors by enhancing autonomy and positive work attitudes, thereby facilitating organizational innovation and performance enhancement. Therefore, we propose the following hypothesis:

**Hypothesis** **1.**
*Empowering leadership positively correlates with employees’ workplace proactivity.*


### 2.2. Mediating Role of Team Job Crafting Between Empowering Leadership and Workplace Proactivity

Team job crafting is the collaborative process by which groups of employees identify strategies to restructure work tasks in alignment with common work objectives. In team job crafting, members collaborate to leverage their efforts and competencies. Consequently, they amplify structural and social job resources, enhance challenging job demands, and reduce hindering job demands ([51]). Previous researchers have defined team job crafting as the collaborative efforts of team members to enhance structural and social job resources, address challenging job demands, and diminish hindering job demands. Although a growing body of research investigates the significance of job crafting in organizations, most studies examine how individuals design their jobs ([17]). Research on team job crafting, where individuals collaboratively design and reshape their jobs with others, remains limited ([68]). Consequently, this aspect also serves as this study’s focal point. The emergence of globalized work, virtual teams, and self-managed teams has significantly heightened the complexity and flexibility of professional tasks. The work environment has increasingly become dynamic due to the incessantly evolving roles, tasks, and projects ([33]). Within the framework of the JD-R (job demands–resources) theory, the principal objective of job-reshaping behaviors is to enhance job proactivity and avert health impairments ([40]). In response to pressing demands and opportunities in the workplace, managers increasingly emphasize the importance of employees adapting and proactively modifying their work ([39]). Employees’ proactive behavior in the workplace is crucial for organizational effectiveness ([71]). Proactive individuals can swiftly identify opportunities, initiate actions, demonstrate proactivity, and actively pursue challenges. They comprehensively delineate their roles and consistently redefine their responsibilities to incorporate new goals and tasks ([30]). Job crafting is a bottom-up approach for redesigning work, wherein employees autonomously institute changes based on the characteristics of their tasks. As organizations experience rapid changes in their structure, processes, and functions, employees require a more profound comprehension of how their job roles and responsibilities contribute to achieving organizational objectives ([29]). Team job crafting encompasses collaboration among team members to adjust and customize tasks aimed at enhancing team performance and work experience ([39]). Research indicates that empowering leadership is particularly pertinent to team job crafting ([42]; [48]). Empowering leadership denotes a set of leadership behaviors that emphasize the significance of work tasks, inspire enthusiasm and motivation for the work, and enhance awareness of initiating and regulating actions ([25]). Effective communication between exemplary leaders and team members may substantially enhance employees’ engagement in team job crafting ([59]). The conservation of resources theory ([34]) posits that supportive leader behaviors, such as empowerment, and positive personal characteristics, such as workplace proactivity, are resources that individuals pursue and can result in positive outcomes. Team job crafting may function as a mechanism for transmitting the influence of empowering leadership ([44]). Empowering leadership is typically linked to behaviors that enhance autonomy and offer support ([60]). Team job crafting emphasizes collaborative teamwork to adjust the work environment, thereby enhancing cohesion among team members and increasing their motivation to collectively pursue positivity and innovation in their work ([43]). Providing adequate support, fostering trust, and promoting empowering leadership are essential for enhancing team member self-efficacy. This leadership style cultivates team members’ confidence in executing collaborative tasks ([70]). Enhanced employee empowerment fosters increased autonomy in team tasks and facilitates collaborative sharing of work adjustments. This sharing of experiences can progressively stimulate work initiative and instill in team members a sense that their collective efforts significantly benefit the entire team and work environment ([40]). An atmosphere of empowerment and collaboration bolsters individual self-efficacy and establishes a team culture that promotes mutual growth and success ([49]). Therefore, we propose the following hypothesis:

**Hypothesis** **2.**
*Team job crafting mediates the relationship between empowering leadership and workplace proactivity.*


### 2.3. Mediating Role of Psychological Safety Between Empowering Leadership and Workplace Proactivity

Empowering leadership is one of the most extensively researched leadership styles. This style is characterized by leaders who exhibit behaviors that promote employee autonomy while stimulating active participation among the workforce ([63]). Empowering leadership is generally linked to open communication and dialogue ([41]). Team psychological safety promotes employees’ willingness to proactively share ideas and pose questions, thereby fostering open communication among team members ([3]). Through open communication and dialogue, it facilitates the transmission of empowering leadership messages, inspiring employee engagement and innovation. This, in turn, enhances creativity and proactive behavior in the workplace ([72]). Simultaneously, empowering leadership establishes a stable and secure work environment for employees ([45]). A stable work environment encourages employees to embrace risks and generate new ideas, essential for fostering innovation and proactive behavior. Employees with elevated psychological safety are inclined to articulate their ideas and pursue essential resources to enhance performance, free from the fear of negative feedback. This encourages employees to actively engage in creative and proactive activities ([14]). Employees with a proactive personality fulfill their designated tasks within the organization and propel the task completion process with heightened creativity and efficiency, thereby enhancing overall work efficiency ([17]). Empowering leadership can elevate employees’ trust in their leaders ([69]). Psychological safety fosters trust among team members ([22]). In the context of empowering leadership, employees tend to cultivate trust in both the leader and the team members’ shared goals. This trust inspires employees to engage more actively in their work, thereby exhibiting enhanced creativity and proactivity ([74]). Empowering leadership prioritizes employees’ development and growth ([49]). Therefore, we propose the following hypothesis:

**Hypothesis** **3.**
*Psychological safety mediates the relationship between empowering leadership and workplace proactivity.*


### 2.4. Moderating Effect of Access to Knowledge via ICT on the Relationship Between Team Job Crafting and Workplace Proactivity

As time progresses, organizations increasingly adopt novel working methods to adapt to the flexible and adaptable work environment influenced by emerging technologies ([20]). The pervasive adoption of digital technology is fundamentally transforming the dynamics of work, life, and organizations ([58]). ICT, particularly smartphones, empowers employees to maintain continuous connectivity to their work from any location. As a pivotal catalyst in the modern economy, ICT fosters innovation and enhances productivity ([65]). “Access to knowledge via ICT” indicates that organizations equip employees with the resources necessary to acquire knowledge using ICT. ICT encompasses various technologies and tools, such as the internet, computers, smartphones, and software applications, that facilitate the acquisition, storage, processing, and transmission of information ([20]). In recent decades, several organizations have embraced trust-based and empowered human resource management (HRM) practices. This entails providing and sharing autonomy, fostering flexible work environments, and leveraging ICT to provide information, all intended to motivate employees ([57]). The new ways of working facilitate knowledge sharing among employees, both within and beyond the organization, by providing electronic tools and ICT ([19]). Reinforcing HRM practices enhances employees’ intrinsic motivation. However, further empirical evidence is necessary to ascertain how specific practices in authorized HRM, such as professional autonomy and access to knowledge via ICT, can effectively stimulate workplace proactivity ([20]). Digital ICT comprises a range of tools available for selection and utilization in the work process ([50]). Team members can effortlessly exchange information, experiences, and perspectives through ICT. Information sharing can enhance team collaboration, thereby facilitating more effective job crafting ([58]). ICT enables remote work, thus enabling team members to independently select their work locations and schedules. This flexibility may enhance employees’ autonomy, thereby increasing their propensity to demonstrate proactivity ([40]). The organization’s provision of advanced ICT tools enhances innovation and collaboration, enabling employees to share information, exchange ideas, and promote team cooperation and innovation ([66]). When team members can efficiently and promptly employ ICT in their work processes, they will more readily access diverse types of knowledge, including work-related information, recent advancements, and industry trends ([36]). Therefore, this study predicts that knowledge acquisition via ICT will be essential for fostering positive team reshaping and encouraging employees to exhibit proactivity in the workplace. Therefore, we propose the following hypothesis:

**Hypothesis** **4.**
*Access to knowledge via ICT reinforces the relationship between team job crafting and workplace proactivity.*


Figure 1 shows the research model.

## 3. Research Methodology

### 3.1. Sample and Procedure

This study’s data were obtained from leaders and employees of seven small and medium-sized IT companies in China. The participants comprised individuals in job roles related to internet development, management, data analysis, and other IT-related fields. Regarding geographical distribution, this study focused on regions that represent the IT industry, including the capitals of Beijing, Guangdong, Jiangsu, and Shanghai. The distribution of the questionnaires was conducted with the prior consent of both leaders and employees, ensuring that our research adheres to ethical standards and upholds participant rights. The employees chosen to complete the questionnaire were required to have been employed at their current company for at least six months ([61]). To mitigate potential common method bias and enhance data validity, this study adopted a multi-source, time-lagged design involving two distinct participant groups: 510 full-time employees (subordinates) and their respective 74 team leaders. Data were systematically gathered across three phases, with a two-week interval between each wave, spanning from 11 September to 8 November 2023. Employees evaluated their leaders’ empowering leadership practices, whereas team leaders assessed team job crafting behaviors within their teams in phase 1. Employees reported their perceptions of psychological safety at the individual level and their access to organizational knowledge via ICT tools in phase 2. Team leaders evaluated their subordinates’ workplace proactivity, ensuring an objective assessment of behavioral outcomes in phase 3.

Responses were linked via unique team codes (e.g., Team01-Employee001) without collecting personally identifiable information to maintain anonymity and enable cross-level analysis. Additionally, collaboration with each company’s human resources department facilitated effective coordination with team leaders, resulting in a response rate of over 85% across all phases. This structured approach minimized common method variance and aligned with ethical standards by safeguarding participant confidentiality. Given that all participants are Chinese, we utilized the back translation method to ensure the consistency and rigor of the translated language. The survey encompassed empowering leadership, workplace proactivity, team job crafting, psychological safety, and access to knowledge via ICT. The analytical tools utilized were SPSS 27.0, Amos 28.0, and HLM 6.08.

This study employed SPSS 27.0 to conduct descriptive statistics on the basic demographic characteristics of valid samples. Of the 74 team leaders, 54.1% (N = 40) were male, whereas 45.9% (N = 34) were female. Most leaders were between 41 and 50 years old, accounting for 44.6% (N = 33) of the sample. Leaders above 50 years old constituted 14.9% (N = 11), whereas those aged between 20 and 30 years accounted for 4.1% (N = 3). Regarding the leaders’ educational background, 68.9% (N = 51) of leaders held bachelor’s degrees. Regarding ICT usage, 4.1% (N = 3) of leaders allocated less than 70% of their working hours to ICT, 51.3% (N = 38) dedicated 70–85%, and 44.6% (N = 33) exceeded 85%. Regarding team size, 78.4% (N = 58) managed teams comprising six to nine members, while 2.7% (N = 2) managed teams exceeding 10 members. Within the cohort of 510 subordinates, 51.4% were identified as male (N = 262), whereas 48.6% were female (N = 248). Regarding age distribution, 39.0% (N = 199) were aged between 20 and 30 years, while 44.9% (N = 229) were aged between 31 and 40 years. In contrast, individuals above the age of 50 accounted for only 3.4% (N = 17). Regarding educational qualifications, 58.8% (N = 300) held a bachelor’s degree. Regarding ICT usage, 33.9% (N = 173) devoted less than 70% of their daily working hours to ICT, whereas 49.0% (N = 250) allocated 70% to 85% of their daily working time to ICT. Additionally, 17.1% (N = 87) reported employing ICT for more than 85% of their daily work hours. Table 1 describes the demographic characteristics of the participants.

### 3.2. Measures

For each employee’s evaluation of empowering leadership, we utilized a 12-item scale developed by [1] ([1]). The leader’s evaluation of employee team job crafting was measured using a five-item scale developed by [67] ([67]). This study utilized the employee self-reported Psychological Safety Five-Item Scale developed by [13] ([13]). To evaluate employees’ access to knowledge via ICT, we utilized an employee self-reported four-item scale developed by [18] ([18]). We adopted a 13-item scale developed by [56] ([56]) to measure leaders’ evaluation of employees’ workplace proactivity. All items were measured using a five-point Likert-type scale, which ranged from 1 (strongly disagree) to 5 (strongly agree).

### 3.3. Analytic Strategy

To examine the multi-level hypotheses, we employed hierarchical linear models (HLM) ([11]). Furthermore, the analysis procedure for multi-level mediation follows the guidelines established by [75] ([75]). We employed full maximum likelihood to estimate the parameters. The level 1 variables were applied to the technique “group-mean centering and adding the group mean at level 2,” whereas grand-mean centering was used for the level 2 variables. Based on [75] ([75]), the mediating effect of team job crafting on the relationship between empowering leadership and workplace proactivity was classified as a 2-2-1 model, indicating a cross-level mediation effect with a superior mediator. However, the mediation of psychological safety in the relationship between empowering leadership and workplace proactivity was identified as a 2-1-1 model, signifying a cross-level mediation effect with a lower mediator. All mediation analyses were performed in accordance with [9]’s ([9]) recommendations. A cross-level interaction term (TJC WP) was established and integrated into the model to test the hypothesis concerning the moderating effect of access to knowledge via ICT.

## 4. Data Analysis and Results

### 4.1. Preliminary Analyses

Table 2 illustrates the reliability and validity scores for all variables. The Cronbach’s alpha values exceeded the threshold of 0.70 ([28]), thus confirming all variables’ internal consistency. Similarly, the AVE and CR values exceeded 0.50 and 0.70, respectively ([32]). Consequently, all variables’ reliability and validity scores were deemed acceptable for subsequent analysis.

Table 3 presents the variables’ mean values as follows: empowering leadership = 3.847, team job crafting = 3.298, psychological safety = 3.930, access to knowledge via ICT = 3.498, and workplace proactivity = 3.298. Furthermore, the standard deviation values for all variables were within the normal range. Additionally, a binary correlation was identified among the research variables in the assumed direction; therefore, the study data were deemed suitable for further analysis. Meanwhile, the square root of the AVE values, which is displayed along the diagonal, exceeded the correlation values, thereby exhibiting discriminant validity.

### 4.2. Hypothesis Tests

In Model 1 (see Table 4), we added control variables based on the null model 1 and discovered that these control variables had no significant effect on the dependent variable. In Model 2, we incorporated the team variable of empowering leadership, yielding a regression coefficient of 1.318 (*p* < 0.001) for empowering leadership’s effect on workplace proactivity. The result indicated that empowering leadership significantly impacted workplace proactivity, thus supporting Hypothesis 1. Upon the integration of team job crafting (mediator variable) into Model 3, the positive correlation between empowering leadership and workplace proactivity weakened (r = 0.887, *p* < 0.001), whereas team job crafting positively correlated with workplace proactivity (r = 0.263, *p* < 0.001). This demonstrates that team job crafting partially mediates the relationship between empowering leadership and workplace proactivity, thus supporting Hypothesis 2.

Model 4 demonstrates a significant positive effect of empowering leadership on psychological safety, indicated by a coefficient of 0.734 (*p* < 0.001). Upon incorporation of psychological safety (mediating variable) into Model 6, the positive correlation between empowering leadership and workplace proactivity weakened (from 1.318, *p* < 0.001 in Model 2 to 1.181, *p* < 0.001). In contrast, psychological safety positively correlated with workplace proactivity (r = 0.189, *p* < 0.001. This observation demonstrated that psychological safety partially mediates the relationship between empowering leadership and workplace proactivity, thereby supporting Hypothesis 3.

To evaluate the moderating effect of “access to knowledge via ICT,” we incorporated a new interaction term (team job crafting * access to knowledge via ICT) into our model. Initially, we incorporated the moderating variable “access to knowledge via ICT” into Model 7; however, the results indicated that “access to knowledge via ICT” did not significantly impact workplace proactivity (r = 0.003, *p* = 0.96). Subsequently, we incorporated “team job crafting” into Model 8, based on Model 7. After this addition, we included the interaction term (team job crafting * access to knowledge via ICT) in Model 9. The results demonstrated that this interaction term (team job crafting * access to knowledge via ICT) significantly impacted workplace proactivity (r = 0.150, *p* < 0.01), thereby supporting Hypothesis 4.

Table 5 illustrates that the mediating value for the empowering leadership–team job crafting–workplace proactivity is 0.493, with Z = 7.825 and *p* < 0.001, thus supporting the presence of a mediating effect. For the empowering leadership–psychological safety–workplace proactivity, the mediating value is 0.141, with Z = 4.134 and *p* < 0.001, thereby also supporting the presence of a mediating effect. Therefore, Hypotheses 2 and 3 are supported.

Figure 2 illustrates the presence of access to knowledge via ICT in the correlation between job crafting and workplace proactivity. Regardless of the level of access to knowledge via ICT, an increase in workplace proactivity is evident as team job crafting increases. The impact of team job crafting on workplace proactivity is more pronounced when access to knowledge via ICT is high. This indicates that increased access to knowledge via ICT enhances the positive effects of team job crafting on workplace proactivity. Conversely, the enhancement in workplace proactivity arising from access to knowledge via ICT is less significant when team job crafting is low. However, this difference becomes increasingly evident as team job crafting increases, demonstrating that access to knowledge via ICT is more effective in contexts with abundant resources.

## 5. Results and Discussion

This study, grounded in the job demands–resources (JD-R) theoretical framework, systematically elucidates the “black box” of how empowering leadership fosters workplace proactivity from a multi-level perspective. The findings reveal that empowering leadership directly enhances individual proactive behaviors and exerts a cross-level transmission effect by influencing team-level job-crafting practices and fostering an atmosphere of psychological safety. This dual-pathway validation transcends the conventional one-dimensional interpretation of leadership effects by incorporating the dynamic interplay between individual and team mechanisms into the analysis.

Notably, the moderating effect of access to knowledge via ICT clarifies the limitations of technological empowerment in a digital context. Equipping teams with efficient ICT infrastructure significantly enhances the positive influence of leadership on job crafting, resulting in a more sustainable cycle of proactive behavior.

This study develops a three-dimensional model encompassing leadership behavior, team processes, and the technological context. It offers a novel analytical framework to comprehend the factors influencing proactivity in complex organizational environments. It establishes a robust empirical foundation for subsequent theoretical development and organizational interventions.

### 5.1. Theoretical Implications

Firstly, this study explores the impacts of enhanced leadership capabilities on employee proactivity in the workplace. Grounded in previous research, this study demonstrates that empowering leadership significantly affects employee proactivity in the workplace ([31]; [52]; [8]). Particularly, this observation is evident when employees are assigned high-diligence responsibilities ([63]). Despite the increasing academic focus on empowering leadership, limited studies have investigated the correlation between empowering leadership and workplace proactivity ([52]; [47]; [63]). In this context, this study has developed a novel theoretical framework that broadens the research scope of empowering leadership. Moreover, it offers new insights into the factors influencing employee proactivity in the workplace.

Secondly, to elucidate the mediating mechanisms between empowering leadership and employee workplace proactivity, we present team job crafting and psychological safety as variables. We constructed a dual mediation model to examine whether team job crafting and psychological safety can modulate the relationship between enhanced leadership capabilities and employee workplace proactivity. Job crafting, an emergent bottom-up approach to work design, has recently garnered immense attention ([71]). Research on job crafting predominantly focuses on the individual dimension, with limited consideration of the team dimension ([53]). According to the JD-R (job demands–resources) theory, job crafting behaviors primarily aim to improve workplace proactivity and prevent health impairments ([40]). This study’s multi-level model posits that the empowerment of leadership behaviors, through the mediating factors of team job crafting and psychological safety, can inspire team members to actively participate in team job crafting, thereby enhancing employees’ positive orientation towards their work. This emphasizes the team-level mediating processes, which enhances our comprehension of the mechanisms that propel workplace proactivity. It enriches the current research regarding the impact of empowering leadership on employee workplace proactivity.

Thirdly, by examining the impact of access to knowledge via ICT on team job crafting and workplace proactivity, the theoretical framework offers a more comprehensive understanding of the features of modern work environments. Access to knowledge via ICT does not directly enhance individual workplace proactivity; however, its positive impact on workplace proactivity is evident when integrated with job crafting. This confirms that access to knowledge via ICT empowers team members to more effectively reshape their work content and structure, thereby enhancing their work proactivity. This finding significantly enhances theory because it illustrates how team behaviors, such as information collection, indirectly influence employee motivation by transforming the work environment (i.e., job crafting). Consequently, it establishes a theoretical basis for comprehending the impact of ICT on employee autonomy, thereby enhancing the theoretical framework for analyzing employee proactivity in relation to innovation.

### 5.2. Practical Implications

This study underscores the significant positive impact of empowering leaders on workplace positivity. Our findings indicate that leaders can enhance workplace positivity by granting employees greater autonomy, decision-making power, and increased responsibility. This strategy cultivates a conducive environment that fosters initiative and innovation among employees. In such an environment, they are motivated to maximize their efforts, achieve their work goals, enhance their performance, and enhance workplace motivation. By cultivating this leadership style, organizations can diminish employees’ reliance on top-level decision-making and incentivize them to assume responsibility and proactively confront challenges.

Furthermore, this study presents team job crafting and team psychological safety as mediating variables. It demonstrates that empowered leaders can inspire employees by leveraging the mediating effects of team job crafting and psychological safety. This provides a robust framework for firms to design and optimize leadership development strategies. Organizations can prioritize developing leaders’ empowering leadership style to enhance their ability to facilitate team job crafting and psychological safety, thereby fostering heightened employee motivation. Simultaneously, prioritizing the establishment of an environment conducive to teamwork reinvention and psychological safety in project management and team development may enhance teamwork effectiveness and foster innovative initiatives. Because teamwork is a crucial factor in the success of IT projects, this study offers practical guidance for optimizing teamwork and fostering psychological safety within teams.

The results of the moderated effects indicate that while “access to knowledge via ICT” does not directly enhance work proactivity, its integration with “team job crafting” facilitates the application and translation of this knowledge into innovative and adaptive behaviors within the team. This finding emphasizes that managers promote employee initiative and innovation by effectively integrating technological resources and job design in the creation of work environments and team structures. Effectively facilitating employee access to knowledge resources and fostering a supportive team environment will enhance organizational effectiveness. Moreover, it implies that enhancing ICT access to knowledge can facilitate more effective team job crafting, thereby enhancing employee initiative and overall productivity.

## 6. Limitations and Future Research Direction

This study has certain limitations. Firstly, due to time and budget constraints, we employed a time-lagged data collection method, which entailed gathering questionnaire data over a specific period. Consequently, this method restricts our findings to a specific period, thus precluding any assertions of causality. Therefore, future research should use longitudinal studies to evaluate the model and explore the long-term effects of empowered leadership on employee motivation in greater depth, thereby achieving more accurate findings.

Secondly, this study’s data were sourced from IT companies in economically advanced regions of China; hence, it exclusively represents a single industry and culture. Therefore, our findings may be subject to cultural bias or country-specific organizational practices. To address this limitation, future studies should collect data from multicultural organizations or diverse demographic perspectives to mitigate cultural prejudice ([35]).

Thirdly, this paragraph discusses the impact of diverse ICT employed by various teams on workplace dynamics. It emphasizes that certain teams effectively utilize ICT to enhance knowledge and initiative. However, other teams may encounter difficulties due to technical barriers or inadequate training. These barriers can undermine the moderating effect of knowledge acquisition via ICT on the correlation between teamwork creation and workplace initiative. This inconsistency implies that the challenges encountered by team members while using ICT may diminish their workplace initiative, thereby impacting ICT’s overall effectiveness as a moderator. In conclusion, this paper recommends that future research should comprehensively address and control for these discrepancies to ensure a more accurate representation of real-world situations. This will enhance the finding’s reliability and applicability.

Finally, during this study’s data collection phase, only leaders were tasked with assessing employee motivation in the workplace. Future research should involve both leaders and employees jointly assessing employee motivation in the workplace, thereby providing a more comprehensive understanding of the assessment’s validity and authenticity.

## Figures and Tables

**Figure 1 behavsci-15-00609-f001:**
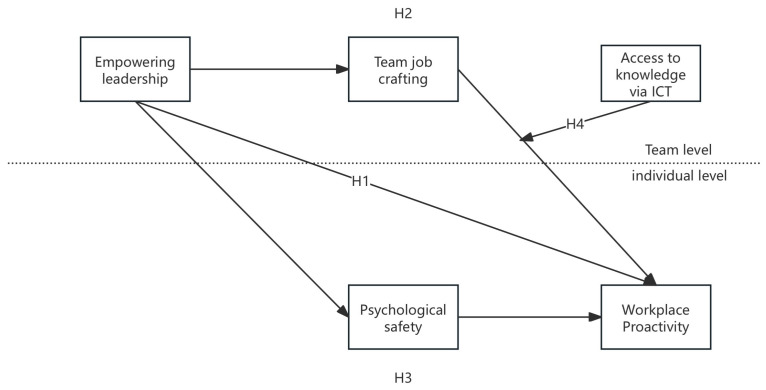
Research model.

**Figure 2 behavsci-15-00609-f002:**
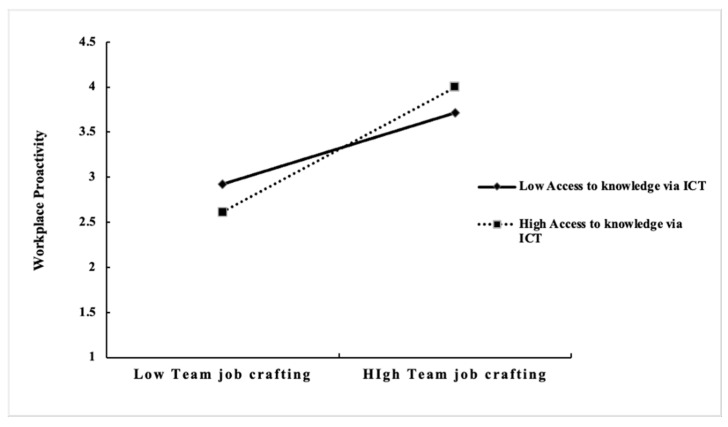
Moderating effect diagram of access to knowledge via ICT on team job crafting—workplace proactivity.

**Table 1 behavsci-15-00609-t001:** Descriptive characteristics of the sample.

Team Leaders		Employees Demographic	
Characteristic	Percent (Number)	Characteristic	Percent (Number)
Gender		Gender	
Male	54.1% (40)	Male	51.4% (262)
Female	45.9% (34)	Female	48.6% (248)
Age		Age	
20–30	4.1% (3)	20–30	39.0% (199)
31–40	36.4% (27)	31–40	44.9% (229)
41–50	44.6% (33)	41–50	12.7% (65)
>50	14.9% (11)	>50	3.4% (17)
Education level		Education level	
Junior College degree or below	24.3% (18)	Junior College degree or below	37.1% (189)
Bachelor’s degree	68.9% (51)	Bachelor’s degree	58.8% (300)
Master’s degree or above	6.8% (5)	Master’s degree or above	4.1% (21)
ICT-usage frequency		ICT-usage frequency	
Below 70%/per day	4.1% (3)	Below 70%/per day	33.9% (173)
70–85%/per day	51.3% (38)	70–85%/per day	49.0% (250)
Above 85%/per day	44.6% (33)	Above 85%/per day	17.1% (87)
Team size			
2–5 people	18.9% (14)		
6–9 people	78.4% (58)		
More than 10 people	2.7% (2)		

**Table 2 behavsci-15-00609-t002:** Scales’ reliability and validity.

Variable	Items	Alpha	Factor Loading	CR	AVE
Empowering leadership	12	0.773	0.687–0.818	0.775	0.534
Team job crafting	5	0.942	0.850–0.944	0.943	0.769
Psychological safety	5	0.832	0.634–0.748	0.833	0.501
Access to knowledge via ICT	4	0.909	0.830–0.840	0.909	0.714
Workplace proactivity	13	0.896	0.809–0.870	0.896	0.685

Note: Alpha = Cronbach’s alpha, AVE = average variance extracted, CR = composite reliability.

**Table 3 behavsci-15-00609-t003:** Means, standard deviations, and correlations of variables studies.

Variable	Mean	SD	1	2	3	4	5
Empowering leadership	3.847	0.556	(0.731)				
Team job crafting	3.298	1.217	0.567 **	(0.877)			
Psychological safety	3.930	0.780	0.317 **	0.236 **	(0.708)		
Access to knowledge via ICT	3.498	1.124	0.201 *	0.102 *	0.115 **	(0.845)	
Workplace proactivity	3.298	0.821	0.550 **	0.450 **	0.387 **	0.021	(0.828)

Note: The square root of AVE is presented along the diagonal. ** *p* < 0.01 * *p* < 0.05.

**Table 4 behavsci-15-00609-t004:** Regression analysis for hypothesis.

	Null Model 1	Model 1	Model 2	Model 3	Null Model 2	Model 4	Model 5	Model 6	Model 7	Model 8	Model 9
Variables	Workplace Proactivity				Psychological Safety		Workplace Proactivity				
Intercept	3.311 ***	3.314 ***	3.311 ***	3.310 ***	3.927 ***	3.926 ***	3.314 ***	3.312 ***	3.313 ***	3.310 ***	3.313 ***
Level 1											
Employee gender		0.028	0.030	0.035		0.065	0.012	0.016	0.029	0.039	0.035
Employee age		0.020	0.027	0.020		0.042	0.010	0.019	0.020	0.009	0.015
Employee education		−0.038	−0.058	−0.058		−0.050	−0.029	−0.047	−0.037	−0.047	−0.041
ICT use in daily work		0.022	0.021	0.025		−0.013	0.023	0.024	0.022	0.029	0.017
Psychological safety							0.233 ***	0.189 ***			
Level 2											
Leader gender		−0.098	−0.111	−0.105		−0.098	−0.078	−0.091	−0.097	−0.111	−0.102
Leader age		−0.135	−0.017	−0.006		0.008	−0.123	−0.018	−0.134	−0.016	−0.017
Leader education		0.183	0.124	0.135		0.316 ***	0.102	0.065	0.183	0.162	0.158
ICT use in daily work		0.249	0.027	0.004		0.073	0.203	0.013	0.249	0.025	0.037
Team size		−0.238	−0.086	−0.054		0.026	−0.224	−0.091	−0.238	−0.050	−0.018
Empowering leadership			1.318 ***	0.887 ***		0.734 ***		1.181 ***			
Team job crafting				0.263 ***						0.628 ***	0.575 ***
Access to knowledge via ICT									0.003	−0.025	−0.011
TJB* ICT										0.150 **	
R (Sigma_squared)	0.388	0.390	0.394	0.395	0.416	0.419	0.381	0.382	0.390	0.393	0.390
U(Tau)	0.277	0.260	0.036	0.024	0.197	0.104	0.197	0.029	0.265	0.060	0.201
Chi-square	448.975 ***	391.162 ***	111.102 ***	97.192 *	309.495 ***	178.785 ***	320.158 ***	103.207 **	397.158 ***	135.720 ***	307.822 ***
Deviance	1095.229	1115.771	1037.165	1033.536	1106.837	1101.563	1091.242	1022.503	1117.279	1057.675	1102.199

Note *** *p* < 0.001, ** *p* < 0.01, * *p* < 0.05; TJB* ICT = team job crafting; * access to knowledge via ICT.

**Table 5 behavsci-15-00609-t005:** Sobel test.

Variable	A	Sa	b	Sb	a*b	Z	*p*
EL-TJB-WP	0.834	0.151	0.592	0.061	0.439	7.825	0.000
EL-PS-WP	0.514	0.153	0.275	0.088	0.141	4.134	0.000

Note: EL = empowering leadership, TJB = team job crafting, WP = workplace proactivity, PS = psychological safety.

## Data Availability

Data will be accessible upon request.

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
