# Peer review of "How Empowering Leadership Drives Proactivity in the Chinese IT Industry: Mediation Through Team Job Crafting and Psychological Safety with ICT Knowledge as a Moderator"

_behavsci, 2025, doi:10.3390/bs15050609_

Round 1
Reviewer 1 Report
Comments and Suggestions for Authors
Dear Author(s),
This is an interesting study; however, i think it requires some revisions to enhance its clarity and impact. Please consider the detailed comments below to improve the quality of your work.
Abstract and Title:
While the abstract provides a comprehensive overview of the study, consider explicitly highlighting the study's contributions—the mediating and moderating effects—to make it more distinctive. I suggest to revise the title to emphasize these effects and the importance of the study.
Introduction:
The authors should better clarify why this is an important research problem, and how the study contributes to the existing literature. I think the authors must improve the discussion about the gap in the literature and to provide a better motivation to the study.
A clear discussion of the research questions in this section is highly recommended.
Additionally, it is suggested to introduce more information and analysis about the Chinese geographical and cultural context, the specificity of the IT industry, and how it is different compared with population and firms from other cultural background and industries.
I suggest to identify the hypotheses in the schematic representation of the conceptual model (figure 1).
Sample and procedure:
As a potential reader of the paper, I would expect to see a detailed table with descriptive statistics.
Results:
The results section is well-organized, with clear tables and figures that align with textual descriptions. However, the table 4 is not indicated previously in the text, even it is not properly contextualized and explained.
Comments on the Quality of English LanguageProofread the manuscript for minor issues and improve sentence flow in certain sections.
Author Response
Dear Reviewer 1,
We are grateful for your thoughtful and constructive feedback on our manuscript. We sincerely appreciate the time and effort you invested in reviewing our work.
Based on your suggestions, we have meticulously revised the manuscript to improve clarity, theoretical framing, and contextual specificity. Specifically, we have:
- Enhanced the Introduction by elucidating the theoretical gap and practical relevance of the study.
- Refined the articulation of our research questions and clarified the study’s novel contribution to the literature on empowering leadership in digital contexts.
- Reinforced the contextual background of China’s IT industry to highlight its distinctiveness from other nations.
- Revised the title and abstract to emphasize the mediating and moderating mechanisms examined.
- Integrated the research hypotheses into Figure 1 to enhance conceptual clarity.
- Added a comprehensive table of descriptive statistics (now Table 1) to enhance transparency in the sample and procedure section.
- Accurately contextualized and elucidated the Sobel test results in the Results section (now Table 5).
All changes have been indicated in the revised manuscript. We hope that these revisions adequately address your concerns and enhance the quality and clarity of our paper. For point-by-point revisions, please refer to the detailed responses below.
Once again, we appreciate your insightful feedback.
Sincerely,
Juhee Hahn
On behalf of all co-authors
Point-by-point response to Comments and Suggestions for Authors |
Comments 1: Abstract and Title: While the abstract provides a comprehensive overview of the study, consider explicitly highlighting the study's contributions—the mediating and moderating effects—to make it more distinctive. I suggest to revise the title to emphasize these effects and the importance of the study |
Response 1: I appreciate your insightful suggestion. In response, we have revised both the title and the abstract to emphasize the study’s contributions, specifically the mediating and moderating effects examined in our model. The updated title now emphasizes the theoretical mechanisms and the digital context in which they occur, effectively conveying the study’s focus. The updated abstract (lines 9–29, page 1) now explicitly outlines the mediating functions of team job crafting and psychological safety, as well as the moderating effect of access to knowledge via ICT. These revisions seek to enhance the clarity and distinctiveness of our study’s contributions. |
Comments 2: Introduction: The authors should better clarify why this is an important research problem, and how the study contributes to the existing literature. I think the authors must improve the discussion about the gap in the literature and to provide a better motivation to the study. |
Response 2: Thank you for highlighting this critical point. We have amended the relevant text to more explicitly articulate the significance of the research problem. Please refer to the revised paragraph 2 on Page 3, Lines 102–107. |
Comments 3: Introduction: A clear discussion of the research questions in this section is highly recommended. |
Response 3: We appreciate this recommendation. We have restructured the section to present three research questions more distinctly. Please refer to Page 4, Paragraph 2, Lines 116–131 |
Comments 4: Introduction: Additionally, it is suggested to introduce more information and analysis about the Chinese geographical and cultural context, the specificity of the IT industry, and how it is different compared with population and firms from other cultural background and industries. |
Response 4: Thank you for this insightful comment. We have incorporated contextual information regarding the Chinese IT industry, including its scale, work culture (e.g., “996” work schedule), and comparison with Western IT firms. Moreover, we discussed the hierarchical and high-pressure cultural milieu that influences leadership and employee behaviors in China, thereby enriching the contextual foundation of our research. Please refer to the updated Page 2, Paragraph 2, Lines 46–62.
|
Comments 5: I suggest to identify the hypotheses in the schematic representation of the conceptual model (figure 1). |
Response 5: We appreciate the proposal. Figure 1 has been revised to incorporate the relevant hypothesis labels, elucidating the theoretical relationships depicted in the model and enhancing its readability. Please refer to the revised Figure 1 on Page 10, Line 323. |
Comments 6: Sample and procedure: As a potential reader of the paper, I would expect to see a detailed table with descriptive statistics. |
Response 6: Thank you for your useful suggestion. In response, we have included a detailed Table 1 featuring the descriptive statistics. This table enhances the clarity and transparency of our sample characteristics and data structure. Please refer to Page 11, Table 1, Line 378.
|
Comments 7: The results section is well-organized, with clear tables and figures that align with textual descriptions. However, the table 4 is not indicated previously in the text, even it is not properly contextualized and explained.
|
Response 7: |
5. Additional clarifications |
We appreciate the reviewers’ insightful comments and recommendations. Apart from the specific revisions noted above, we have meticulously reviewed the entire manuscript to enhance clarity, coherence, and consistency. All grammatical issues and language expressions have been enhanced to elevate the quality of English. We invite any further recommendations and are willing to revise the manuscript if necessary. |
Reviewer 2 Report
Comments and Suggestions for Authors
First, I want to congratulate you on a well written paper, on a very interesting and actual topic, the relationship between empowering leadership and workplace productivity. However, I have a few comments to make:
- the introduction section of a paper should generally present the context of your research, it's general aim or purpose, its novelty and the paper's structure. Your introduction section, in my opinion, is mainly a theoretical discussion of the concepts used in your paper, which is redundant since the same theoretical concepts are discussed in section 2. For example, lines 46-57 from the introduction and lines 93-109 from section two discuss, pretty much the same concepts, only using different words. I would suggest changing the introduction.
- Fig 1 - I would also add the hypothesis on the lines connecting the variables in the theoretical model. At the same time, please make sure that the theoretical model reflects all the relationships between your variables, as you have stated them in the hypothesis development section. For example, H1 deals with the direct relationship between empowering leadership and workplace productivity, variables which are not directly connected in your theoretical model figure.
- it's not clear which exactly is your sample. Did you apply the questionnaire only to the 510 subordinates or to the 74 team leaders as well?
Author Response
Dear Reviewer 2,
We sincerely appreciate your time in reviewing our manuscript and for your insightful and constructive comments. We genuinely value your insights, which have significantly enhanced the clarity and overall quality of the paper. We have provided a detailed point-by-point response in the accompanying revision table, with all relevant revisions distinctly highlighted red in the revised manuscript.
In light of your feedback, we have revised the following texts: First, we appreciate your feedback regarding the redundancy between the introduction and the theoretical background section. Based on your recommendation, we have thoroughly revised the introduction by eliminating redundant theoretical discussions (e.g., definitions of key constructs). Rather, we have prioritized the research context (Chinese IT companies), the general aim and novelty of the study, and the overall structure of the paper. The omitted content is now exclusively addressed in Section 2. These revisions can be found on pages 1–3 of the revised manuscript.
Second, we appreciate your identification of the inconsistencies between the theoretical model and the stated hypotheses. We have revised Figure 1 by (1) distinctly marking each path with its corresponding hypothesis (e.g., H1, H2, etc.) and (2) updating the diagram to reflect all hypothesized relationships, including the direct link between empowering leadership and employee proactivity as stated in H1. The revised Figure 1 is now thoroughly aligned with the hypothesis development section.
Finally, we appreciate your comment regarding the ambiguity in the sample description. In the methodology section, we have clarified that the questionnaire was administered to both 510 subordinates and 74 team leaders. This clarification has been added to Chapter 3 (Research Methodology). Thank you once more for your insightful comments and constructive suggestions. Please do not hesitate to let us know if additional clarifications or revisions are required.
Sincerely,
Juhee Hahn
On behalf of all co-authors
Point-by-point response to Comments and Suggestions for Authors |
Comments 1: the introduction section of a paper should generally present the context of your research, it's general aim or purpose, its novelty and the paper's structure. Your introduction section, in my opinion, is mainly a theoretical discussion of the concepts used in your paper, which is redundant since the same theoretical concepts are discussed in section 2. For example, lines 46-57 from the introduction and lines 93-109 from section two discuss, pretty much the same concepts, only using different words. I would suggest changing the introduction. |
Response 1: I greatly appreciate your constructive suggestion. Following your advice, I have revised the introduction section to eliminate redundancy. Specifically, I have deleted overlapping theoretical discussions and streamlined the conceptual content. To enhance the value of the introduction and more accurately reflect the research context, I have incorporated a description of the distinctive characteristics of Chinese IT companies in the second and third paragraphs on page 2. Moreover, I have deleted the redundant definitions and more explicitly linked the discussion to the research question in the fourth paragraph of page 2 (line 22) and the first paragraph of page 3 (lines 72–74). These revisions intend to concentrate the introduction on the study's background, purpose, and contributions, rather than restating theoretical content elaborated in Section 2. We hope that these revisions have enhanced the clarity and relevance of the introduction section. |
Comments 2: Fig 1 - I would also add the hypothesis on the lines connecting the variables in the theoretical model. At the same time, please make sure that the theoretical model reflects all the relationships between your variables, as you have stated them in the hypothesis development section. For example, H1 deals with the direct relationship between empowering leadership and workplace productivity, variables which are not directly connected in your theoretical model figure. |
Response 2: We appreciate the proposal. Figure 1 has been revised to incorporate the corresponding hypothesis labels, elucidating the theoretical relationships depicted in the model and enhancing its readability. Please refer to the revised Figure 1 on Page 10, Line 323. |
Comments 3: it's not clear which exactly is your sample. Did you apply the questionnaire only to the 510 subordinates or to the 74 team leaders as well? |
Response 3: We thank you for highlighting this ambiguity. In the revised manuscript, we have clarified the sample composition and data collection procedure in the Methodology section (pp. 10-11, lines 335-356). |
3. Additional clarifications |
In addition to the revisions addressing the reviewer’s main concerns, we have also thoroughly proofread the manuscript to enhance language clarity and consistency. To enhance its readability and presentation quality, we have made minor corrections in wording and formatting throughout the manuscript.
|